# Hospital Admissions Related to Infections and Disorders of the Skin and Subcutaneous Tissue in England and Wales

**DOI:** 10.3390/healthcare10102028

**Published:** 2022-10-14

**Authors:** Mohammed Samannodi

**Affiliations:** Department of Medicine, College of Medicine, Umm Al-Qura University, Makkah 24382, Saudi Arabia; mssamannodi@uqu.edu.sa; Tel.: +966-125501000

**Keywords:** admissions, disorders, England, infections, skin, subcutaneous, wales

## Abstract

Objectives: To investigate hospital admissions in England and Wales due to infections and diseases of the skin and subcutaneous tissue. Methods: Data from the Patient Episode Database for Wales (PEDW) and the Hospital Episode Statistics (HES) database in England for the years between April 1999 and April 2020 were used in this study. Using all the relevant diagnosis codes (L00–L99), hospital admissions related to various skin infections and diseases of the subcutaneous tissue were identified. Results: Hospital admissions for all causes increased overall by 78.8%, from 276,464 in 1999 to 494,433 in 2020, representing an increase in hospital admission rate of 56.1% (from 530.23 (95% CI 528.26–532.20) in 1999 to 827.92 (95% CI 825.62–830.22) per 100,000 people in 2020, *p* ≤ 0.05). The most prevalent diagnoses were disorders of the skin’s appendages, infections of the skin and subcutaneous tissue, and other disorders of the skin and subcutaneous tissue. Nearly half of all hospital admissions were for males and for patients between the ages of 15 and 59. In 2020, the hospital admission rate for males increased by 60.2%, from 540.16 (95% CI 537.32–543.01) per 100,000 people in 1999 to 865.10 (95% CI 861.76–868.44) in 2020. From 520.75 (95% CI 518.02–523.48) in 1999 to 791.03 (95% CI 787.86–794.19) in 2020, the hospital admission rate for females grew by 51.9%. Conclusion: Hospital admission due to infections and disorders of the skin and subcutaneous tissue increased during the past two decades in England and Wales. Further studies are needed to explore the risk factors associated with infections and disorders of the skin and subcutaneous tissue complications, and its associated admissions.

## 1. Introduction

The most frequent reason for patients to consult with general practitioners (GPs) is a skin disease, which is a serious public health concern [1]. According to a previous cross-sectional large-scale study in the United Kingdom (UK) in 1976 on 2180 adults, 55% of them had some form of skin condition, and more than 22% of them were considered to be severe enough to warrant medical attention by members of the primary care team, such as a community pharmacist, nurse, or general practitioner [2]. Approximately 6.0% and 10%, respectively, of outpatient referrals and a GP’s workload are related to skin diseases [1]. Skin diseases account for 12–23% of all symptom-based requests for advice from community pharmacists [3].

There are about 3000 acute and chronic skin diseases known; they affect people of all ages and socioeconomic status [4]. Acne, atopic dermatitis, and eczema were the most prevalent skin conditions in Europe [5]. In Europe, the three most prevalent skin diseases were atopic dermatitis or eczema, acne, and fungal skin infections [5]. In the UK now, 60% of people have skin disorders or have had them in the past [6]. Numerous studies have shown that skin conditions have a major impact on quality of life, productivity, and mental health [7,8]. About 70% of British citizens who have scars or skin conditions say they have an influence on their confidence [6]. Skin problems are the third most common industrial disorder and a leading cause of time missed from work, which has a high economic impact on both society and the person [1]. Skin diseases ranked as the fourth most common non-fatal disability cause worldwide [9].

The majority of hospitalized patients suffer from skin diseases [10]. In England and Wales, between 1999 and 2019, the percentage of viral infections that resulted in skin and mucous membrane lesions requiring hospitalization increased by 51.9% [11]. Comparing the main disease classifications in England and Wales, skin diseases are the most frequent reason for general practice consultations [12]. Clinicians, researchers, and policymakers will be better able to improve outcomes and lessen the financial and social burdens that skin diseases place on societies, the healthcare system, and individuals, by properly allocating resources when they are informed about the burden of skin diseases among inpatients. There have been studies conducted in the past in the UK that looked at the admissions profile for various acute and chronic illnesses [13,14,15,16,17,18,19,20,21,22,23], but none that looked at trends in infections and disorders of the skin and subcutaneous tissue. Therefore, the purpose of this study was to investigate hospital admissions in England and Wales due to infections and diseases of the skin and subcutaneous tissue.

## 2. Methods

### 2.1. Study Sources and the Population

The Hospital Episode Statistics (HES) database in England [24] and the Patient Episode Database for Wales (PEDW) for the time period between April 1999 and April 2020 [25] were used in this study (on the population level), which store publicly accessible data. The HES and PEDW databases provide information on hospital admissions for people with infections and diseases of the skin and subcutaneous tissue from all age groups, which are separated into four categories: those under 15 years, those between 15 and 59 years, those between 60 and 74 years, and those aged 75 years and over. The International Statistical Classification of Diseases and Related Health Problems (ICD-10) Fifth Edition, used by the National Health Service (NHS) to categorize diseases and other health conditions, was used to identify hospital admissions. All hospital admissions associated with various skin infections and diseases of the subcutaneous tissue were identified using all the relevant diagnosis codes (L00–L99). Data from patients whose primary diagnosis for admission did not match one of these codes were excluded. All hospital admissions, readmission, transfer, outpatient visits, and accident and emergency (A&E) visits made at all National Health Service (NHS) trusts and any independent sector sponsored by NHS trusts are documented in the HES and PEDW databases. Data are reported stratified by sex for each hospital admission episode.

Starting in 1999/2000, information on hospital admissions in England and Wales is available. Patient demographics, clinical diagnosis, treatments, and length of stay are all included in the data. The yearly hospital admission rate between 1999 and 2020 was determined using Office for National Statistics (ONS) midyear population data.

### 2.2. Statistical Analysis

Using the finished consultant episodes of admission divided by the mid-year population, hospital admission rates with 95% confidence intervals (CIs) were determined. To compare the variation in hospital admission rates between 1999 and 2020, a chi-squared test was applied. All analyses were conducted using SPSS version 27 (IBM Corp, Armonk, NY, USA).

## 3. Results

Hospital admissions for all causes increased overall by 78.8% from 276,464 in 1999 to 494,433 in 2020, representing an increase in hospital admission rate of 56.1% (from 530.23 (95% CI 528.26–532.20) in 1999 to 827.92 (95% CI 825.62–830.22) per 100,000 people in 2020, *p* < 0.05). The most common indications were infections of the skin and subcutaneous tissue, other disorders of the skin and subcutaneous tissue, and disorders of skin appendages, which accounted for 45.5%, 25.3%, and 15.8%, respectively (Table 1).

During the study duration, a tremendous increase in hospital admissions rate was seen in infections of the skin and subcutaneous tissue and “urticaria and erythema”, 1.45-fold and 1.41-fold, respectively. Furthermore, the hospital admissions rate for radiation-related disorders of the skin and subcutaneous tissue, bullous disorders, dermatitis and eczema, and other disorders of the skin and subcutaneous tissue was increased by 96.0%, 71.9%, 33.2%, and 27.1%, respectively. However, the hospital admissions rate for disorders of skin appendages and papulosquamous disorders decreased by 30.8% and 23.0%, respectively (Figure 1 and Table 2).

In terms of age differences for hospital admission, those between the ages of 15 and 59 years made up 50.5% of the total, followed by those 75 years and over with 21.7%, those between the ages of 60 and 74 years with 20.0%, and those less than 15 years with 7.9%. From 276.25 (95%CI 272.98–279.52) in 1999 to 290.89 (95%CI 287.67–294.11) in 2020, hospital admission rates for patients under the age of 15 years increased by 5.3%. From 487.47 (95%CI 485.04–489.91) in 1999 to 652.68 (95%CI 649.99–655.37) in 2020, the rate of hospital admission for patients aged 15 to 59 years increased by 33.9%. From 720.31 (95%CI 714.01–726.60) in 1999 to 1147.74 (95%CI 1140.91–1154.56) in 2020, hospital admission rates for patients aged 60 to 74 years increased by 59.3%. Patients 75 years old and over experienced a 1.10-fold rise in hospital admission rates, going from 1177.84 (95%CI 1167.15–1188.53) in 1999 to 2475.15 (95%CI 2461.76–2488.54) in 2020 per 100,000 people (Figure 2).

During the study period, there were 7,608,889 hospital admission episodes reported in England and Wales. In total, 3,887,209 hospital admission episodes, or a mean of 185,105 admissions per year, were of males, who made up 51.1% of the total number of hospital admissions. In 2020, the hospital admission rate for males increased by 60.2%, from 540.16 (95% CI 537.32–543.01) in 1999 to 865.10 (95% CI 861.76–868.44) per 100,000 people in 2020. From 520.75 (95% CI 518.02–523.48) in 1999 to 791.03 (95% CI 787.86–794.19) in 2020, the hospital admission rate for females grew by 51.9% (Figure 3).

### 3.1. Diseases of the Skin and Subcutaneous Tissue Admission Rate by Sex

Most hospital admission rates for males were greater than for females. This comprises skin and subcutaneous tissue infections, papulosquamous disorders, bullous disorders, radiation-related skin and subcutaneous tissue disorders, and disorders of skin appendages (Figure 4). However, compared to males, females had greater hospital admission rates for dermatitis and eczema, urticaria and erythema, and other conditions affecting the skin and subcutaneous tissue (Figure 4).

### 3.2. Diseases of the Skin and Subcutaneous Tissue Admission Rate by Age Group

Multiple types of hospital admissions, such as infections of the skin and subcutaneous tissue, bullous disorders, radiation-related disorders of the skin and subcutaneous tissue, and other disorders of the skin and subcutaneous tissue, were observed to be directly related to age (more common among the age group of 75 years and above). However, the age groups of those under 15 years, over 75 years, between 60 and 74 years, and between 15 and 59 years had higher rates of hospital admissions for “dermatitis and eczema” and “urticaria and erythema”, respectively. The age groups of 15–59 years, 60–74 years, 75 years and above, and under 15 years experienced higher rates of hospital admissions due to disorders of skin appendages. Hospitalizations for papulosquamous disorders were more frequent in the following age groups: 60–74 years, 15–59 years, 75 years and older, and under 15 years, respectively (Figure 5).

## 4. Discussion

This study presents hospitalization data at the national level for individuals with infections and disorders affecting the skin and subcutaneous tissue. The following are the study’s primary findings: (1) infections of the skin and subcutaneous tissue, other disorders of the skin and subcutaneous tissue, and disorders of skin appendages were the most common indications; (2) males and patients aged 15 to 59 years made up nearly half of all hospital admissions; and (3) the hospital admission rate among males increased higher than that of females during the study period. During the study period, the hospital admission rate for infections and disorders of the skin and subcutaneous tissue in England and Wales increased by 56.1%. This study confirmed the findings of a previous study in the United States (US), which reported that there has been an increase in the number of visits to emergency rooms and hospitalizations for skin infections and that 13.0% of hospitalization among adults was related to skin diseases [10]. Hospitalizations for skin infections are on the rise, and this has a significant financial impact on healthcare resources [26]. One of the main contributing factors to this noticeable increase in admission rates is the increase in the population itself. The ONS reported that the total population in England and Wales increased by 14.5% between 1999 and 2020 [27]. Previous research suggested that the improvement in life expectancy for both sexes and the high annual migration rates between 2005 and 2020 could be linked to population growth [28]. A growing prevalence of comorbidities, such as diabetes, the spread of multidrug-resistant pathogens, and an aging population are additional factors influencing the increase in bacterial skin infections [10].

This study identified that the most common indications were infections of the skin and subcutaneous tissue, other disorders of the skin and subcutaneous tissue, and disorders of skin appendages, which accounted for 45.5%, 25.3%, and 15.8%, respectively. Similar results from other studies conducted in New Zealand, Australia, and the US revealed that hospitalization for skin and soft tissue infections is rising and more frequent than for other dermatological conditions [29,30,31]. O’Sullivan et al. reported that the incidence of serious skin infections almost doubled between 1990 and 2007 in Australia [31]. Miller et al. found that in outpatient and inpatient settings, skin and soft tissue infections are steadily rising in the US [30]. Thean et al. examined the hospital admissions for skin and soft tissue infections in a population with endemic scabies in Fiji and found that the incidence of this type of infection is affected by age (incidence was highest at the extremes of age) and ethnicity [32]. Schofield et al. reported that the most frequent diagnostic category that patients presented with in primary care was skin infections, which is also the most prevalent cause of consultation in general practice in England and Wales [12]. There are multiple factors that contributed to the increase in prevalence of skin and soft tissue infections and its associated hospitalization. The sharp increase in community-associated methicillin-resistant *Staphylococcus aureus* skin infections is one of the main contributing factors to the increase in skin and soft tissue infections [33,34,35]. Cellulitis is another common type of skin infections that lead to hospitalization among the pediatric population [36]. Previous literature has reported that infections among children become more prevalent due to several factors, such as poorer general hygiene, delay in seeking medical attention, and experiencing frequent injuries [31]. Socioeconomic deprivation is another contributing factor that increase the possibility of serious skin infections [31]. Among the elderly population, there are many disposing factors, such the presence of comorbidities (e.g., diabetes mellitus) that increase the possibility of them getting infected with skin and soft tissue infections [32].

Our study found that the age group 15–59 years accounted for 50.5% of the total number of hospital admissions. The age range 15–59 years contributed to 27.1% of all admissions related to infectious and parasitic diseases in England and Wales, according to findings from a previous research by Sweiss et al. [23]. Similarly, a study by Naser et al. found that the age range of 15 to 59 years accounted for 27.2% of hospital admissions for viral infections characterized by skin and mucous membrane lesions [11]. The clinical range of infections of the skin and soft tissues, particularly those of the head and neck, is quite diverse. These infections can either be mild and localized or invasive and life-threatening. The clinical presentation and outcome of skin infections are influenced by a number of variables. These variables include the virulence of the infecting organism(s), the host’s immunologic state, and the presence of infection-promoting circumstances. The relative proximity of these superficial infections to important structures in head and neck infections frequently presents potential clinical challenges [37].

In our study, males contributed to 51.1% of the total number of hospital admissions. The hospital admission rate among males increased by 60.2% and among females increased by 51.9%. Most of the hospital admission rates were higher among males compared to females. It has been noted that males experience skin infections more frequently than females, which is consistent with our findings [38]. According to the article published by Naser et al., 60% of hospital admissions for viral skin infections were for males [11]. There is a sex difference in the incidence of skin diseases, according to numerous publications in the literature. The mechanisms causing the sex disparity in skin diseases are still unknown. Different hypotheses have partially explained these findings. It might be related to the rapidly rising male population in the UK, as reported in 2020 [28]. These variations may be caused by variations in skin structure, the impact of sex hormones, ethnic background, sociocultural behavior, and environmental factors [38].

According to previous research, there may be a correlation between less access to dermatologists who see outpatient patients and a higher risk of skin diseases hospitalization [10]. Given the high prevalence of hospitalization for skin diseases, expanding access to dermatological consulting services is advised. Dermatologists, primary-care doctors, infectious disease specialists, and other outpatient medical staff can also try to lessen the inpatient burden of skin infections by addressing risk factors such as ulcers, fungal infections, and excoriations, as well as by adhering to clinical guidelines for empiric antibiotics to prevent outpatient treatment failures [39,40,41].

This study has multiple strong points. To the best of my knowledge, this is the first study to explore hospital admissions in England and Wales due to all types of infections and diseases of the skin and subcutaneous tissue without restricting the study population to a specific age group or specific skin condition. The main limitation of this study is the study design that presented the data at the population level and not at the individual level, which restricted the ability to define other risk factors associated with the increase or decrease in hospital admissions due to infections and diseases of the skin and subcutaneous tissue. This study reported data on emergency, readmission, and elective hospital admissions; therefore, the results should be carefully assessed, since this may have led to an overestimation of the presented admission rates. Other limitations include a lack of data on ethnicity, rural/urban residence, and gender at the age-group level.

## 5. Conclusions

Hospital admission due to infections and disorders of the skin and subcutaneous tissue increased during the past two decades in England and Wales. Further studies are needed to explore the risk factors associated with infections and disorders of the skin and subcutaneous tissue complications, and its associated admissions.

## Figures and Tables

**Figure 1 healthcare-10-02028-f001:**
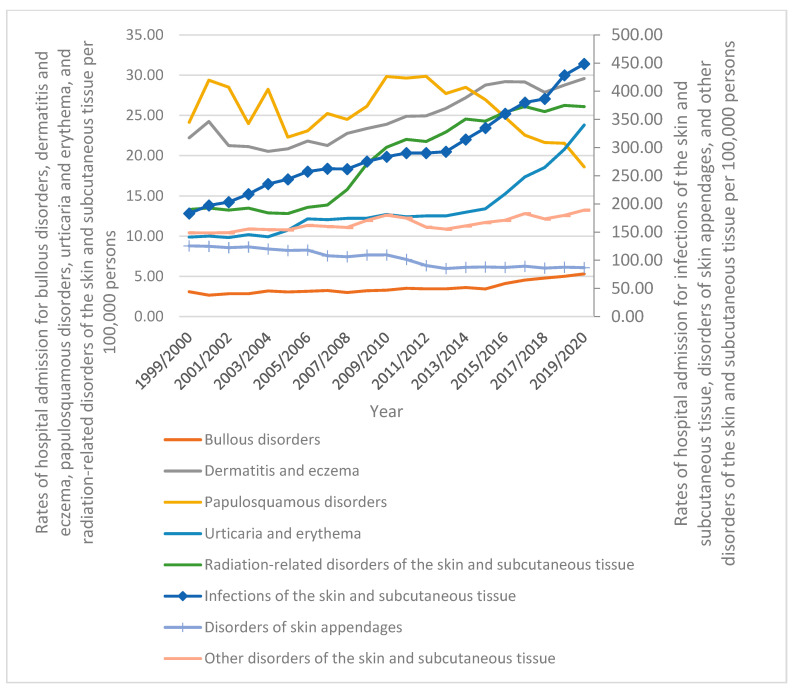
Rates of hospital admission stratified by type between 1999 and 2020.

**Figure 2 healthcare-10-02028-f002:**
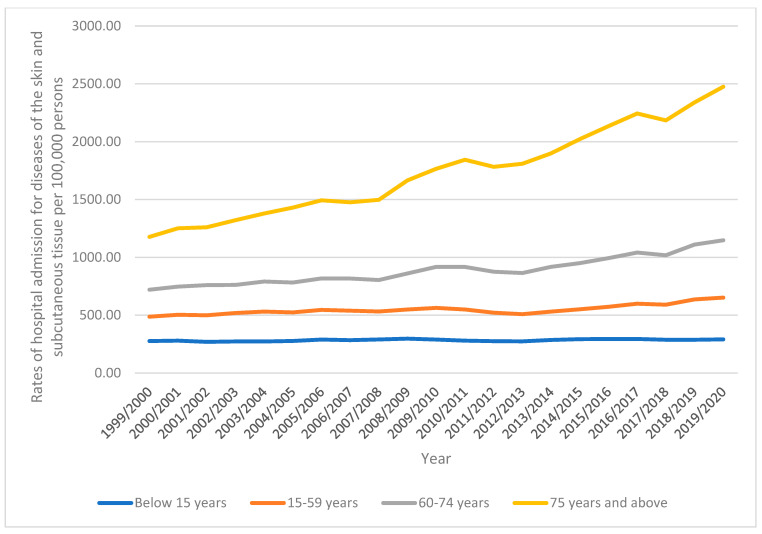
Rates of hospital admission stratified by age group.

**Figure 3 healthcare-10-02028-f003:**
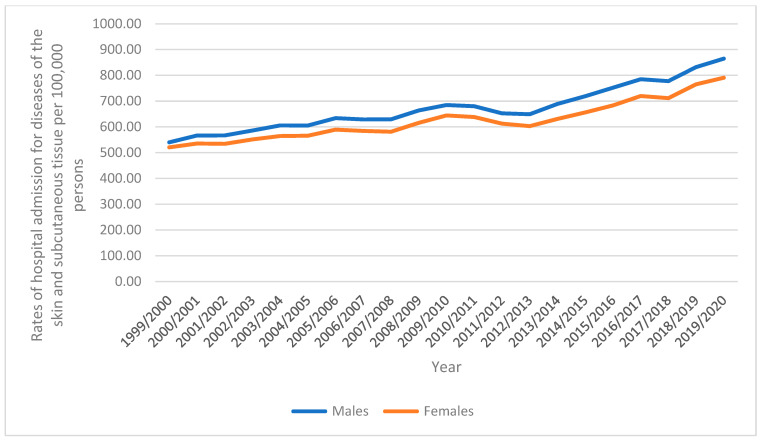
Rates of hospital admission stratified by sex.

**Figure 4 healthcare-10-02028-f004:**
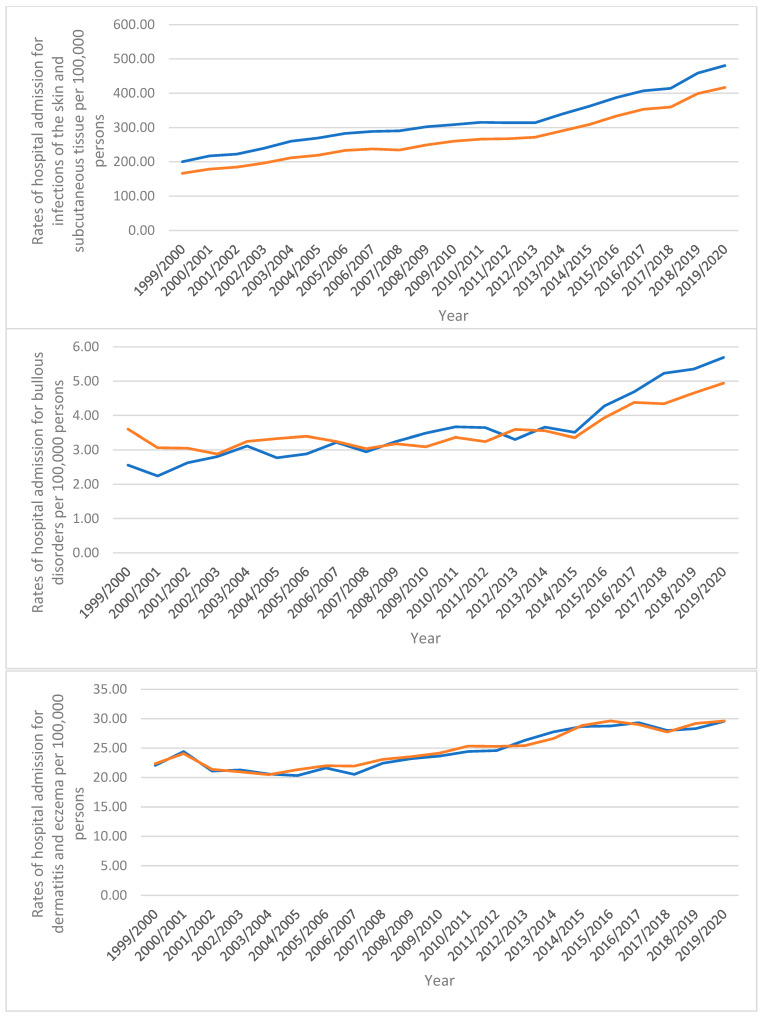
Hospital admission rates stratified by sex.

**Figure 5 healthcare-10-02028-f005:**
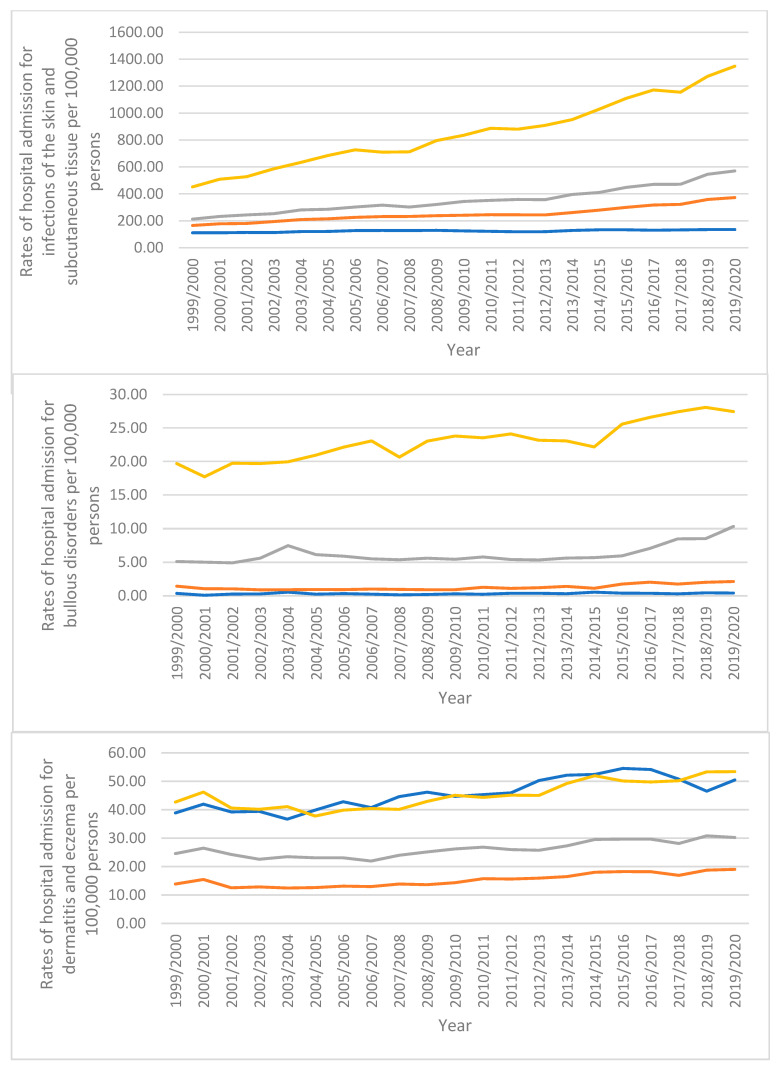
Hospital admission rates stratified by age.

**Table 1 healthcare-10-02028-t001:** Percentage of diseases of the skin and subcutaneous tissue hospital admission from the total number of admissions.

ICD Code	Description	Percentage from Total Number of Admissions
L00–L08	Infections of the skin and subcutaneous tissue (staphylococcal scalded skin syndrome, impetigo, cutaneous abscess, furuncle and carbuncle, cellulitis and acute lymphangitis, acute lymphadenitis, pilonidal cyst and sinus, and other local infections of skin and subcutaneous tissue).	45.5%
L80–L99	Other disorders of the skin and subcutaneous tissue (vitiligo, other disorders of pigmentation, seborrheic keratosis, acanthosis nigricans, corns and callosities, other epidermal thickening, keratoderma, transepidermal elimination disorders, pyoderma gangrenosum, pressure ulcer, atrophic disorders of skin, hypertrophic disorders of skin, granulomatous disorders of skin and subcutaneous tissue, lupus erythematosus, other localized connective tissue disorders, vasculitis limited to skin, and non-pressure chronic ulcer of lower limb)	25.3%
L60–L75	Disorders of skin appendages (nail disorders, nail disorders in diseases classified elsewhere, alopecia areata, androgenic alopecia, other nonscarring hair loss, cicatricial alopecia [scarring hair loss], hair color and hair shaft abnormalities, hypertrichosis, acne, rosacea, follicular cysts of skin and subcutaneous tissue, other follicular disorders, eccrine sweat disorders, and apocrine sweat disorders)	15.8%
L40–L45	Papulosquamous disorders (psoriasis, parapsoriasis, pityriasis rosea, and lichen planu)	3.9%
L20–L30	Dermatitis and eczema (atopic dermatitis, seborrheic dermatitis, diaper dermatitis, allergic contact dermatitis, irritant contact dermatitis, unspecified contact dermatitis, exfoliative dermatitis, dermatitis due to substances taken internally, lichen simplex chronicus and prurigo, and pruritus)	3.8%
L55–L59	Radiation-related disorders of the skin and subcutaneous tissue	3.0%
L50–L54	Urticaria and erythema (exfoliation due to erythematous conditions according to extent of body surface involved, urticaria, erythema multiforme, erythema nodosum, and oher erythematous conditions)	2.1%
L10–L14	Bullous disorders (pemphigus, other acantholytic disorders, and pemphigoid)	0.6%

ICD: International Statistical Classification of Diseases system.

**Table 2 healthcare-10-02028-t002:** Percentage change in hospital admission rates from 1999 to 2020 in England and Wales.

Diseases	Rate of Diseases in 1999 per 100,000 Persons (95% CI)	Rate of Diseases in 2020 per 100,000 Persons (95% CI)	Percentage Change from 1999 to 2020
Infections of the skin and subcutaneous tissue	183.14 (181.97–184.30)	448.51 (446.81–450.20)	144.9%
Bullous disorders	3.09 (2.94–3.24)	5.32 (5.13–5.50)	71.9%
Dermatitis and eczema	22.21 (21.81–22.62)	29.59 (29.15–30.02)	33.2%
Papulosquamous disorders	24.14 (23.72–24.56)	18.60 (18.25–18.94)	−23.0%
Urticaria and erythema	9.88 (9.61–10.15)	23.80 (23.41–24.19)	140.9%
Radiation-related disorders of the skin and subcutaneous tissue	13.31 (13.00–13.63)	26.09 (25.68–26.50)	96.0%
Disorders of skin appendages	125.66 (124.70–126.62)	86.97 (86.23–87.72)	−30.8%
Other disorders of the skin and subcutaneous tissue	148.80 (147.75–149.85)	189.05 (187.95–190.15)	27.1%

## Data Availability

Publicly available datasets were analyzed in this study. This data can be found here: http://content.digital.nhs.uk/hes (accessed on 13 July 2022), http://www.infoandstats.wales.nhs.uk/page.cfm?pid=41010&orgid=869 (accessed on 13 July 2022).

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
