# Peer review of "Hospital Admissions Related to Infections and Disorders of the Skin and Subcutaneous Tissue in England and Wales"

_healthcare, 2022, doi:10.3390/healthcare10102028_

Round 1
Reviewer 1 Report
The manuscript describe very well the trends in hospital admissions related to the skin disorders. However, you should reconsider if it is an ecological study or only a descriptive or cross-sectional study about the epidemiology surveillance. I also suggest to reduce the number of graphs included in figure 4 and 5 because the trends are similar; and to include only the graphs were you found a difference in the trends.
Author Response
Reviewer 1:
The manuscript describes very well the trends in hospital admissions related to the skin disorders. However, you should reconsider if it is an ecological study or only a descriptive or cross-sectional study about the epidemiology surveillance. I also suggest to reduce the number of graphs included in figure 4 and 5 because the trends are similar; and to include only the graphs were you found a difference in the trends.
- I would like to thank the reviewer for the time and efforts in reviewing our manuscript. I have now clarified the study design which is ecological study design on the population level, see line 69. Regarding the figures, I preferred to present all hospital admissions figures as the reader maybe interested in examining the trend either stratified by age or gender for specific skin and subcutaneous infection or disorder. Figure 4 presents the tends stratified by gender for each skin condition and Figure 5 present the trends stratified by age for each skin condition.
Reviewer 2 Report
The manuscript by Samannodi presents an ecological study on the hospital admissions in England and Wales that were related to skin and subcutaneous tissue diseases, including infections. In this study the author collected publicly available data from the National Health Service from 1999 to 2020 and assessed the prevalence of different types of skin diseases over time, having stratified the results tendencies by sex and age group.
The main strength of this manuscript relies on its originality, constituting the first research study to present epidemiological data on hospital admissions related to skin diseases in the United Kingdom. It is also nicely written, requiring a minor English revisions. Its main limitations are the lack of information regarding data collection, as well as the discussion of results.
Firstly, regarding data collection, the author should mention whether exclusion criteria were considered. For example, did the author consider only cases from the public sector healthcare or from the private sector also? Did the author take into consideration patient transfers? Did the author consider patients who changed gender during the assessed time period? These may be obvious questions for those who are familiar with ICD-10 and with HES and PEDW databases, but for those who are unfamiliar with these programs a clarification into the inclusion/exclusion criteria is warranted. Secondly, I suggest that the author provide some examples of the skin diseases categorized under “other disorders” throughout the manuscript, as it may possibly confuse some readers. Finally, keeping in mind the evolution of the meaning of the concept of “sex” and “gender”, I suggest the author to use only the term “sex” throughout the text (in the present version both terms are found);
The discussion of results seems to have been done somewhat superficially. For most of the length of the Discussion section, the author presented several studies that showed similar tendencies to his own study. However, few reasons are given for the reported tendencies, which needs to be improved to make the manuscript more robust.
I also raise some minor issues:
Lines (L) 15-17: I suggest that these disease categories be presented in a descending order of percentage; this will help readers organize the information;
L 17: I suggest to replace “...for males and patients between...” with “...for males and for patients between...”;
L28: Wouldn’t it be better if this would be rephrased as “...for patients to consult with general practitioners (GPs)...”?
L30: please provide more details regarding the information taken from reference [2] – in which period did these subjects (55%) had skin conditions? At some point during their lives? During a specific time period?
L31: I suggest to replace “...severe or moderately severe enough...” with “...severe enough...”;
Figure 1: why are there two YY axis captions?
L125-126: Surely the values are switched in the sentence “...increased by 60.2%, from 865.10 125 (95% CI 861.76 - 868.44) per 100,000 people in 1999 to 540.16 (95% CI 537.32 - 543.01) in 126 2020”. Please revise;
L184: If a study was published prior to the present manuscript, it should be the other way around – the present paper confirms the previous USA study. Please revise;
L243: The authors mentions that there are multiple strengths for this manuscript. However, only one is mentioned. Please revise;
As a final note – considering the large number of graphs in this manuscript, my suggestion would be to decrease their size and/or to shorten the number of data points in the XX axis.
Author Response
Reviewer 2:
The manuscript by Samannodi presents an ecological study on the hospital admissions in England and Wales that were related to skin and subcutaneous tissue diseases, including infections. In this study the author collected publicly available data from the National Health Service from 1999 to 2020 and assessed the prevalence of different types of skin diseases over time, having stratified the results tendencies by sex and age group.The main strength of this manuscript relies on its originality, constituting the first research study to present epidemiological data on hospital admissions related to skin diseases in the United Kingdom. It is also nicely written, requiring minor English revisions. Its main limitations are the lack of information regarding data collection, as well as the discussion of results.
- I would like to thank the reviewer for the time and efforts in reviewing our manuscript. I have now provided further information about data collection and databases. In addition, I have now checked the language of the manuscript and revised our discussion section.
Firstly, regarding data collection, the author should mention whether exclusion criteria were considered. For example, did the author consider only cases from the public sector healthcare or from the private sector also?
- Thank you for this valuable comment. Its mentioned in the method section, lines 76-80 that “All hospital admissions, outpatient visits, and Accident and Emergency (A&E) visits made at all National Health Service (NHS) trusts and any independent sector sponsored by NHS trusts are documented in the HES and PEDW databases”. I included all the data related to infections and diseases of the skin and subcutaneous tissue, which was identified using the ICD-10 codes L00-L99, see lines 76-78. I have now highlighted that data from patients whose primary diagnosis for admission did not match one of these codes were excluded, see lines 78-79.
Did the author take into consideration patient transfers?
- Yes, the reported data related to patient transfer. The HES and PEDW databases document all patients’ data including admissions, outpatient visits, and Accident and Emergency (A&E) visits made at all National Health Service (NHS) trusts and any independent sector sponsored by NHS trusts in England and Wales. I have now highlighted this further in the method section, lines 78-80.
Did the author consider patients who changed gender during the assessed time period? These may be obvious questions for those who are familiar with ICD-10 and with HES and PEDW databases, but for those who are unfamiliar with these programs a clarification into the inclusion/exclusion criteria is warranted.
- Yes, the reported data include all admissions and re-admissions of all patients stratified by gender. So any patient changed gender will be recorded with the new gender in the new episode. I have now highlighted this in the method, see lines 82-83.
Secondly, I suggest that the author provide some examples of the skin diseases categorized under “other disorders” throughout the manuscript, as it may possibly confuse some readers.
- Thank you for this valuable comment. I have now added all disorders of the skin and subcutaneous tissue under this category in Table 1, beside the codes “L80-L99”, see page 3.
Finally, keeping in mind the evolution of the meaning of the concept of “sex” and “gender”, I suggest the author to use only the term “sex” throughout the text (in the present version both terms are found);
- Thank you for this comment, I have now replaced the term gender with sex, based on the reviewer comment.
The discussion of results seems to have been done somewhat superficially. For most of the length of the Discussion section, the author presented several studies that showed similar tendencies to his own study. However, few reasons are given for the reported tendencies, which needs to be improved to make the manuscript more robust.
- Thank you for this comment, I have now addressed the reviewer comment and added more interpretation to the study findings in the discussion, see lines 221-232.
I also raise some minor issues:
Lines (L) 15-17: I suggest that these disease categories be presented in a descending order of percentage; this will help readers organize the information;
- Thank you for this comment, I have now presented disease categories in a descending order of percentage, see Table 1.
L 17: I suggest to replace “...for males and patients between...” with “...for males and for patients between...”;
- Thank you for this comment, I have now addressed this comment, see line 17.
L28: Wouldn’t it be better if this would be rephrased as “...for patients to consult with general practitioners (GPs)...”?
- Thank you for this comment, I have now addressed this comment, see lines 28-29.
L30: please provide more details regarding the information taken from reference [2] – in which period did these subjects (55%) had skin conditions? At some point during their lives? During a specific time period?
- Thank you for this comment. I have now highlighted that this was a cross-sectional large scale study conducted in 1976 on 2180 adults, see lines 30-34.
L31: I suggest to replace “...severe or moderately severe enough...” with “...severe enough...”;
- Thank you for this comment. I have now addressed this comment, see line 33.
Figure 1: why are there two YY axis captions?
- Thank you for this comment. The first YY axis was used to present “Rates of hospital admission for bullous disorders, dermatitis and eczema, papulosquamous disorders, urticaria and erythema, and radiation-related disorders of the skin and subcutaneous tissue per 100,000 persons” and the secondary YY axis was used to present the remaining conditions which are “Rates of hospital admission for infections of the skin and subcutaneous tissue, disorders of skin appendages, and other disorders of the skin and subcutaneous tissue per 100,000 persons”. They were presented separately to make it easier for the reader to distinguish between the rates, which are almost ten times higher than each other, which will make one of them very hard to observe the changes in its associated trends.
L125-126: Surely the values are switched in the sentence “...increased by 60.2%, from 865.10 125 (95% CI 861.76 - 868.44) per 100,000 people in 1999 to 540.16 (95% CI 537.32 - 543.01) in 126 2020”. Please revise;
- Thank you for this comment. I have now corrected this sentence, see lines 133-135.
L184: If a study was published prior to the present manuscript, it should be the other way around – the present paper confirms the previous USA study. Please revise;
- Thank you for this comment. I have now corrected this sentence, see lines 194-196.
L243: The authors mentions that there are multiple strengths for this manuscript. However, only one is mentioned. Please revise;
- Thank you for this comment. I have now corrected this sentence, see lines 270-276.
As a final note – considering the large number of graphs in this manuscript, my suggestion would be to decrease their size and/or to shorten the number of data points in the XX axis.
- Thank you for this comment. I will adapt the size of the graph as per the journal guidelines upon the production of the manuscript.
Reviewer 3 Report
It’s a well written and interesting paper.
I congratulate the authors to their work and recommend the article for publication after minor revisions.
Line 49: increased how much within what time range? Table1: i think it would be easier to read if the order is accordibg to percentage (highest to lowest) instead of icd code Figure 5: the legend should be added again (maybe enough as supplementary table)Author Response
Reviewer 3:
It’s a well written and interesting paper. I congratulate the authors to their work and recommend the article for publication after minor revisions.
- I would like to thank the reviewer for the time and efforts in reviewing our manuscript.
Line 49: increased how much within what time range?
- Thank you for this comment. I have clarified the time range in the manuscript, see line 53.
Table1: i think it would be easier to read if the order is accordibg to percentage (highest to lowest) instead of icd code
- Thank you for this comment, I have now presented disease categories in a descending order of percentage, see Table 1.
Figure 5: the legend should be added again (maybe enough as supplementary table)
- Thank you for this comment, I have now addressed this comment.
Reviewer 4 Report
The manuscript requires clarification for below mentioned comments.
1) Abstract: Line no.18; “In 2020, the hospital admission rate for males increased by 60.2%, from 865.10 (95% CI 861.76 - 18 868.44) per 100,000 people in 1999 to 540.16 (95% CI 537.32 - 543.01) in 2020”, reframe the sentence, authors mentioned that prevalence increased by 60.2% but later mentioned that 865.10 in 1999 and 540.16 in 2020. Sentence is confusing. Same discrepancy observed in line 125-126.
2) Discussion: Line 184-186 (reference 10), give detail findings, how their study supports the present study.
3) The present study has some limitations especially individual information is lacking to support the precise reason of rise in incidence of skin diseases. Further, the reasons for increased prevalence were not completely addressed. It is highly recommended to provide a comparison of similar studies in a tabular format summarising the differences/similarities in various studies.
4) Authors have cited the possible references to support the increasing trend of some parameters. But some parameters showed decreasing trend for. e.g. Rates of hospital admissions for disorders of skin appendages (Figure 4), Rates of hospital admissions for papulosquamous disorders etc., age group data. How authors can explain these observations.
5) Do authors documented the type of diseases falling under each disease category? It will be very informative for readers if this information can be provided in supplementary table.
6) Conclusion: Line 252; “tissue increased raised during the past two decades in England and Wales.”, Increased raised? Reframe sentence appropriately.
Author Response
Reviewer 4:
- I would like to thank the reviewer for the time and efforts in reviewing our manuscript.
The manuscript requires clarification for below mentioned comments.
1) Abstract: Line no.18; “In 2020, the hospital admission rate for males increased by 60.2%, from 865.10 (95% CI 861.76 - 18 868.44) per 100,000 people in 1999 to 540.16 (95% CI 537.32 - 543.01) in 2020”, reframe the sentence, authors mentioned that prevalence increased by 60.2% but later mentioned that 865.10 in 1999 and 540.16 in 2020. Sentence is confusing. Same discrepancy observed in line 125-126.
- Thank you for this comment. This was unintended typo mistake, I have now corrected this sentence, see lines 18-20 and 133-135.
2) Discussion: Line 184-186 (reference 10), give detail findings, how their study supports the present study.
- Thank you for this comment. I have now added further details from this study to highlight the prevalence of hospitalisation due to skin diseases, see line197.
3) The present study has some limitations especially individual information is lacking to support the precise reason of rise in incidence of skin diseases. Further, the reasons for increased prevalence were not completely addressed. It is highly recommended to provide a comparison of similar studies in a tabular format summarising the differences/similarities in various studies.
- Thank you for this comment. I have now addressed the reviewer comment and add a supplementary table to summarise the differences/similarities in various studies compared to my study, (see supplementary material).
4) Authors have cited the possible references to support the increasing trend of some parameters. But some parameters showed decreasing trend for. e.g. Rates of hospital admissions for disorders of skin appendages (Figure 4), Rates of hospital admissions for papulosquamous disorders etc., age group data. How authors can explain these observations.
- Thank you for this comment. Unfortunately, as I mentioned in the limitation section, the publically available data that was used in this study was on the population level, which restricted our ability to identify other risk factors for the increase or decrease in admission rates. However, based on the reviewer comment, I have now highlighted this point further in the limitations section, see lines 271-276.
5) Do authors documented the type of diseases falling under each disease category? It will be very informative for readers if this information can be provided in supplementary table.
- Thank you for this comment. I have now clarified this point in Table 1.
6) Conclusion: Line 252; “tissue increased raised during the past two decades in England and Wales.”, Increased raised? Reframe sentence appropriately.
- Thank you for this comment. I have now addressed this point.
Round 2
Reviewer 2 Report
The author has satisfactorily addressed all raised issues. The manuscript has now sufficient quality for publication.